# Experimental evidence for bipolaron condensation as a mechanism for the metal-insulator transition in rare-earth nickelates

Jacob Shamblin[1,2], Maximilian Heres[3], Haidong Zhou[1], Joshua Sangoro[3], Maik Lang[2], Joerg Neuefeind [4], J.A. Alonso [5] & Steven Johnston [1]

Many-body effects produce deviations from the predictions of conventional band theory in quantum materials, leading to strongly correlated phases with insulating or bad metallic behavior. One example is the rare-earth nickelates $RNiO_3$, which undergo metal-to-insulator transitions (MITs) whose origin is debated. Here, we combine total neutron scattering and broadband dielectric spectroscopy experiments to study and compare carrier dynamics and local crystal structure in $LaNiO_3$ and $NdNiO_3$. We find that the local crystal structure of both materials is distorted in the metallic phase, with slow, thermally activated carrier dynamics at high temperature. We further observe a sharp change in conductivity across the MIT in $NdNiO_3$, accompanied by slight differences in the carrier hopping time. These results suggest that changes in carrier concentration drive the MIT through a polaronic mechanism, where the (bi)polaron liquid freezes into the insulating phase across the MIT temperature.

[1] Department of Physics and Astronomy, The University of Tennessee, Knoxville, TN 37996, USA. [2] Department of Nuclear Engineering, The University of Tennessee, Knoxville, TN 37996, USA. [3] Department of Chemical and Biomolecular Engineering, The University of Tennessee, Knoxville, TN 37996, USA. [4] Chemical and Engineering Materials Division, Spallation Neutron Source, Oak Ridge National Laboratory, Oak Ridge, TN 37831, USA. [5] Instituto de Ciencia de Materiales de Madrid, CSIC, Cantoblanco, E-28049 Madrid, Spain. Correspondence and requests for materials should be addressed to S.J. (email: sjohn145@utk.edu)

Metal-to-insulator transitions (MITs) driven by many-body effects such as strong electron correlations[1] or electron–phonon (e–ph) interactions[2] have attracted considerable attention. In this context, the perovskite rare-earth-element nickelates $RNiO_3$ (R=La, Pr, Nd, etc.)[3,4] have emerged as an important material family in recent years[3–14]. This family of materials exhibits a high-temperature MIT whose transition temperature $T_{MIT}$ can be tuned continuously by varying the radius of the rare-earth ion[3,4], or through external factors such as strain[5,6], reduced dimensionality[15,16], and heterostructure design[15–19]. This electronic transition is concomitant with a long-range structural change, whereby each $O_6$ octahedra expands and contracts around alternating Ni sites following a breathing-type displacement pattern. The $RNiO_3$ systems are therefore ideal candidates for studying MITs and the ways in which they couple to the lattice or vice versa.

One proposal for the MIT is motivated by formal valence counting, which predicts a $d^7$ valence state for the Ni ion, with a single electron occupying the degenerate Ni $3d$ $e_g$ orbitals in the (nearly) cubic metallic phase. This configuration, however, is unstable towards Jahn–Teller distortions or the formation of a mixed Ni valence state, where alternative Ni atoms acquire $d^{7+\delta}$ and $d^{7-\delta}$ valences for sufficiently large Hund's coupling[7]. This charge disproportionation scenario provides one possible mechanism for the MIT[7,8,20], where the breathing lattice distortion reflects the Ni–O bond distances expected for the different Ni valence states.

An alternative view for the MIT is rooted in the fact that the nickelates likely belong to a wider class of "negative charge transfer" materials[10,12,13,21–24]. In such systems, it is energetically favorable for a Ni hole to move to the surrounding ligand oxygen orbitals. The electronic wave function is then best described as a $\alpha|d^7\rangle + \beta|d^8\underline{L}\rangle$ state[10], where $\beta^2 \gg \alpha^2$ and $\underline{L}$ denotes a hole occupying a molecular orbital formed from the surrounding ligand oxygen $2p_\sigma$ orbitals with an $e_g$ symmetry. The fact that $\beta^2 \gg \alpha^2$ implies that the $d^8\underline{L}$ configuration is lower in energy than the $d^7$ in the atomic picture, and thus the system is in the negative charge transfer regime of the Zaanen–Sawatzky–Allen classification scheme[22].

The idea that holes occupy ligand molecular orbitals is important, because this charge configuration couples strongly to the breathing motion of the oxygen atoms, which can result in a sizable e–ph interaction. Several theoretical studies have shown that this can drive a MIT through bond disproportionation[10–13]. In this scenario, the MIT occurs when pairs of ligand holes occupy alternating $O_6$ octahedra, which then compress to maximize (minimize) the hole's delocalization (kinetic energy) between the O and Ni orbitals. The insulating phase corresponds to a crystal structure where the oxygen sublattice has expanded and contracted around alternating Ni sites along the three crystallographic axes, creating two inequivalent Ni sites and a monoclinic phase, consistent with experiments. The Ni site surrounded by the compressed oxygen octahedron has a ($d^8\underline{L}^2$) charge configuration, while the Ni at the center of the expanded octahedron has a ($3d^8$) configuration[10].

The bond-disproportionation mechanism and negative charge transfer classification of the nickelates has gained significant theoretical[10–12] and experimental support in recent years[13,21,24]. However, an unresolved question concerns the nature of the carriers in the bad metallic phase. The key factor in the bond-disproportionation scenario is the strong coupling between the ligand holes and the bond-stretching motion of the oxygen atoms. If such coupling is present, it should also be active in the metallic phase. It is then natural to wonder how these interactions might affect the properties of the carriers in the metallic state, or if they can produce a polaronic charge carrier. For example, one might

expect that the e–ph interaction could generate a local octahedral compression that binds to the ligand hole. Indeed, such a scenario was evidenced in a recent neutron scattering study[25], as well as prior x-ray absorption[26] and μSR studies[27], which all find indications of two different Ni sites in the metallic phase.

In the following, we performed neutron total scattering and broadband dielectric spectroscopy (BDS) measurements to study the local crystal structure in $LaNiO_3$ and $NdNiO_3$ and its relationship with charge dynamics as a function of temperature. These samples were chosen to contrast $LaNiO_3$, which does not undergo an MIT and remains in a high-symmetry rhombohedral phase down to 0 K, with $NdNiO_3$, which has a phase transition from an orthorhombic to a monoclinic phase that coincides with the MIT at $T \approx 200$ K[25]. Our results provide direct evidence that carriers in both materials are indeed small lattice (bi)polarons, where the local breathing distortions bind with the ligand holes, creating a composite quasiparticle with slow dynamics. These results provide a picture of the nickelates where the bad metallic behavior emerges from a polaronic liquid, which freezes into a charge-ordered state at sufficiently low temperatures.

## Results

**Neutron scattering**. To analyze local distortions to the crystal structure, we first converted our neutron total scattering data for $LaNiO_3$ and $NdNiO_3$ into pair distribution functions (PDFs) as shown in Fig. 1. We initially modeled the data for $LaNiO_3$ with the high-symmetry rhombohedral structure (Fig. 1a) with a single Ni site, rather than the lower-symmetry orthorhombic (Fig. 1b) or monoclinic (Fig. 1c) structure, which is characteristic of other rare-earth nickelates. Although this model fits the average structure well (assessed with diffraction, see supplementary Note 1), the poor quality of fit of the small-box refinement in Fig. 1d (blue line, with a weighted residual of $R_w = 0.186$) indicates that this structure is not consistent with the local atomic arrangement in $LaNiO_3$ at any measured temperature (refinement is only shown at 300 K for clarity). The fit is dramatically improved by using the orthorhombic model (Pnma space group) with distorted $NiO_6$ octahedra; however, some peaks are still modeled poorly (green line, $R_w = 0.098$). Further agreement between the refinement and the data is achieved by lowering the symmetry to the monoclinic model, which allows for two distinct Ni sites (magenta line, $R_w = 0.044$). This improvement in the description of the structure is consistent among all temperatures measured (only data at 300 and 100 K are shown).

While both the orthorhombic and monoclinic structures reproduce the local structure reasonably well, the primary discrepancy between the two pertains to the first peak of the PDF, which represents the nearest-neighbor $\langle Ni - O \rangle$ bond (Fig. 1e): the width of this peak is better reproduced when we allow for at least two inequivalent Ni sites. These results indicate that the structure of $LaNiO_3$ has a high degree of symmetry when viewed on long length scales, but also has at least two different types of Ni sites when viewed locally. These two Ni sites reflect the different environments surrounding the Ni atoms once local octahedral distortions form in agreement with ref. [25].

Our results for $NdNiO_3$ show qualitatively similar behavior; however, the difference in fit between the orthorhombic and monoclinic models are much more subtle (Fig. 1f). Although the PDF for metallic $NdNiO_3$ is reproduced reasonably well with the orthorhombic model (blue line, $R_w = 0.095$), the position and intensity of the $\langle Ni - O \rangle$ bond are slightly misrepresented (blue line, Fig. 1d). Correlated thermal motion of the Ni and O atoms might explain this mismatch; however, such motion should have no impact on the position of the peak. Instead, lowering the symmetry to the monoclinic model significantly improves the

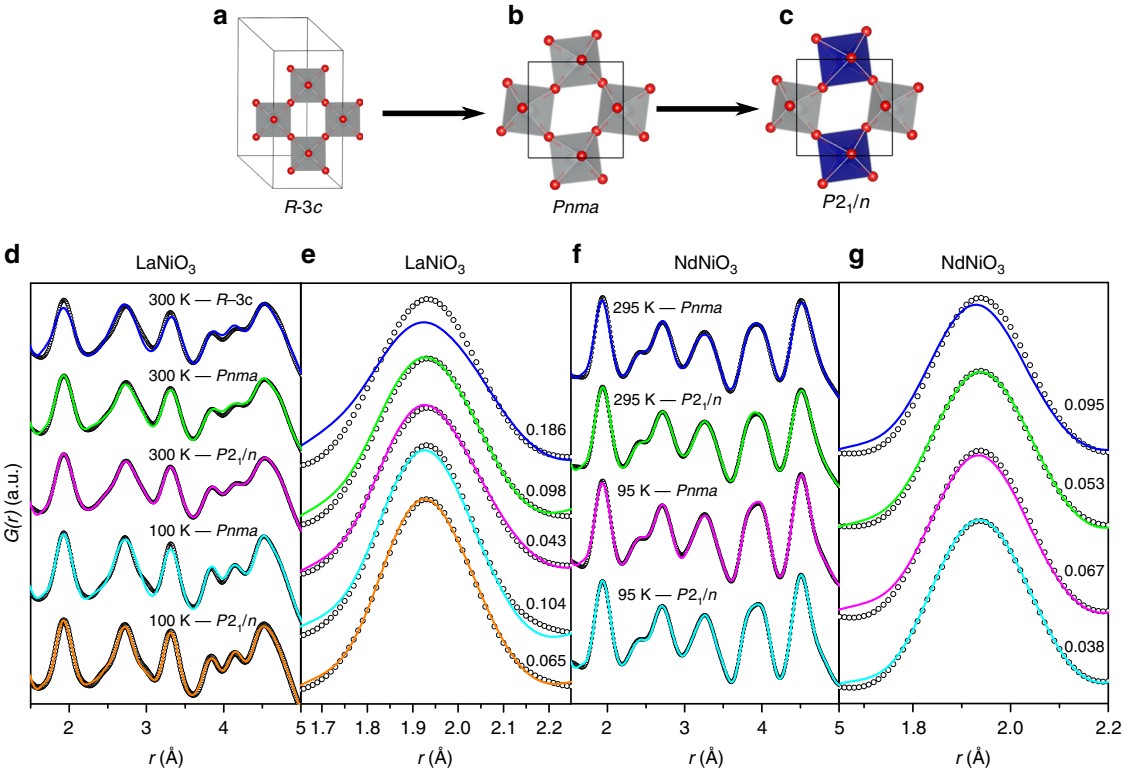

**Fig. 1** Neutron pair distribution function (PDF) and structural results. **a** Rhombohedral structural model with a single $NiO_6$ octahedra (gray polyhedra). **b** Orthorhombic structural model with a single $NiO_6$ octahedra (gray polyhedra). **c** Monoclinic structural model with a contracted (gray polyhedra) and an expanded (blue polyhedra) $NiO_6$ octahedra. **d** Neutron PDFs for $LaNiO_3$ refined with various structural models at 300 and 100 K. **e** Close-up view of refinements in **c** focusing of the first peak in the PDF describing the local $NiO_6$ octahedra. **f** Neutron PDFs for $NdNiO_3$ refined with various structural models at 295 and 95 K. **g** Close-up view of refinements in **c** focusing of the first peak in the PDF describing the local $NiO_6$ octahedra. Open circles in **e** and **f** refer to the experimental PDF whereas solid lines are the refinement. The weighted residual (goodness-of-fit, $R_w$) values for each refinement are shown in **e** and **f**

overall fit (green line in Fig. 1e, $R_w = 0.053$). Although the differences are very slight, this improvement suggests that compressed $NiO_6$ is present in the metallic phase, creating a distribution of long and short Ni–O bonds and the monoclinic model accounts for the inequivalent Ni environments. As expected, the monoclinic model (cyan line, $R_w = 0.0380$) outperforms the orthorhombic model (magenta, $R_w = 0.067$) in $NdNiO_3$ below $T_{MIT}$, with the same improvement occurring in the region of the $NiO_6$ octahedra (Fig. 1c, d).

Although the monoclinic model describes the data better for both samples at all temperatures, we also find clear evidence of a structural transition in $NdNiO_3$ that is present at both short and long length-scales (Fig. 2). Consistent with previous studies[3], our data show that $NdNiO_3$ has an abrupt volumetric expansion at $T_{MIT}$ that is absent in metallic $LaNiO_3$ ($T \approx 195$ K, Fig. 2a). This change is also apparent in the peak intensity of the PDF describing the local structure. As the temperature is lowered from 295 to 195 K in the metallic phase of $NdNiO_3$, the $\langle Ni - O \rangle$ peak height continually grows, as is typical for decreasing thermal motion of the ions (the Debye–Waller effect). However, an apparent discontinuity occurs at $T_{MIT}$, where the peak broadens, and the intensity suddenly decreases. This non-Debye–Waller-like behavior is evidence for a local structural transition concomitant with the average structural response, where expanded and contracted $O_6$ appear along alternating Ni sites across the entire crystal. In contrast, $LaNiO_3$ displays no such transition and the general trend of increasing peak height with decreasing $T$ persists across the entire measured range (except for two outliers at 140 and 260 K). We observe similar contrasting

behavior between $NdNiO_3$ and $LaNiO_3$ in other small-box refinement procedures with an expanded fit range (Supplementary Note 1).

Our neutron data and PDF analysis indicate that the $NiO_6$ octahedra are compressed to varying degrees in both $LaNiO_3$ and $NdNiO_3$ at all temperatures, even in their metallic phases. However, when viewed on long length scales, these distortions are less apparent and the structures are well described by a high-symmetry model. These observations can be reconciled if the lattice distortions are disordered and/or dynamic at high temperatures but "freeze" into a periodic structure below $T_{MIT}$. This suggests a polaronic picture, where the disordered $NiO_6$ octahedra are dynamic quasiparticles, which form a liquid or gas in the metallic phase. Given that the differences in refinements in Fig. 1c, d are tiny, we have employed BDS as a function of temperature to glean further insight into possible dynamic processes that should be present if this picture holds.

**Broadband dielectric spectroscopy.** The frequency range available to the BDS technique (up to $10^{-3}$–$10^{10}$ Hz) provides a competitive advantage to more traditional optical spectroscopies by granting more direct access to the low-frequency processes (e.g., lattice process) that control conduction in a material. The real part of the complex-valued ac conductivity $\sigma'(\nu) + i\sigma''(\nu)$ is plotted in Fig. 3a, b for $LaNiO_3$ and $NdNiO_3$, respectively. At all temperatures, $\sigma'(\nu)$ is constant over a broad frequency range extending up to $\nu \sim 10^5$ Hz and monotonically increases above this range. The frequency-independent portion of the spectrum

represents the dc conduction while $\nu = 1/2\pi\tau_e$ represents a shift from dc to ac behavior.

The data at all measured temperatures can be described well (only the fit at 150 K is shown for clarity) using a continuous-time random walk (CTRW) model $\sigma^*(\nu) = \sigma_{dc}\left[\frac{2\pi i\nu\tau_e}{\ln(1+2\pi i\nu\tau_e)}\right]$, where $\nu$ is the frequency of the ac electric field, $\sigma_{dc}$ is the dc conductivity, and $\tau_e$ is the characteristic carrier hopping time[28]. This model describes the motion of carries hopping in a landscape of randomly varying potential barriers. We have in mind polaronic conduction, where the ligand holes are bound to the local breathing distortion such that coherent motion only occurs when the two move together. In this case, the varying potential barrier arises from variations in the local structural environment, where the barrier height depends on whether or not the neighboring Ni site also hosts a ($d^8\underline{L}$) polaron or a ($d^8\underline{L}^2$) bipolaron, as well as the thermal fluctuations in the lattice distortions. It is important to stress that $\tau_e$ in this model is not the same as the scattering time $\tau$ of the Drude model. In particular, a larger $\tau_e$ describes decreased carrier mobility whereas a larger $\tau$ in the Drude model enhances it. The fact that this model fits our data well in both the metallic and insulating phases indicates similar hopping processes are governing carrier dynamics in both phases.

Using the CTRW model, we extracted the temperature dependence of $\sigma_{dc}$ (Fig. 3c) and $\tau_e$ (Fig. 3d). The dc conductivity for NdNiO$_3$ varies very little at high $T$ but sharply decreases below 195 K. This temperature corresponds to the onset of long-range structural distortions described in Fig. 2 and the MIT reported in previous studies. Conversely, LaNiO$_3$ does not display such a transition as expected. Our BDS experiments are performed on pressed powders and in an air gap geometry (Methods). This setup can produce a value of the conductivity that is reduced by as much as eight orders of magnitude compared with traditional four-probe conductivity measurements[29]. This decrease occurs because no electrical conduction can proceed through the air gap and instead only electrical displacements induced by the alternating electric field are measured. This also results in some sample variation in the overall magnitude of conductivity (Supplementary Notes 2 and 3). We stress, however, that this does not affect the relative trends in conductivity (as evidenced by the clear transition for NdNiO$_3$) or the measured frequency of dynamic processes responding to the applied fields (discussed below). To support this, we have provided a second data set obtained using a different set of samples and varying air gap thicknesses (Supplementary Note 4).

The extracted $\tau_e$ ranges from ≈0.1 to 0.4 μs for NdNiO$_3$ and 0.3 to 0.6 μs for LaNiO$_3$ (Fig. 3c), which are orders of magnitude slower than expected for mobile electrons or holes. We believe that this observation is the consequence of the presence of lattice polarons, where the lattice distortions significantly retard carrier hopping. The $T$-dependence of $\tau_e$ further supports this interpretation. Here, $\tau_e(T)$ varies smoothly for LaNiO$_3$ while there is an abrupt change in slope at $T_{MIT}$ for NdNiO$_3$. More importantly, $\tau_e$ continually decreases with increasing temperature across all regimes in both samples. In general, the dc conductivity is $\sigma_{dc} = ne\mu_e$, where $n$ is the charge carrier concentration, $e$ is the effective charge, and $\mu_e$ is the carrier mobility. The latter is inversely related to the carrier hopping time $\mu_e = \frac{a}{\tau_e E}$, where $a$ is the jump distance and $E$ is the electric field strength. Conductivity is therefore affected equally by the number of charge carriers as well as their mobility. A decrease in dc conductivity with decreasing temperature in the insulating phase of NdNiO$_3$ is consistent with the observed increase in $\tau_e$ (decrease in $\mu_e$). It is surprising, however, that this thermally activated hopping also persists in the metallic phases of both samples. We also note that we can convert our $\tau_e$ values to carrier mobilities with knowledge of the jump

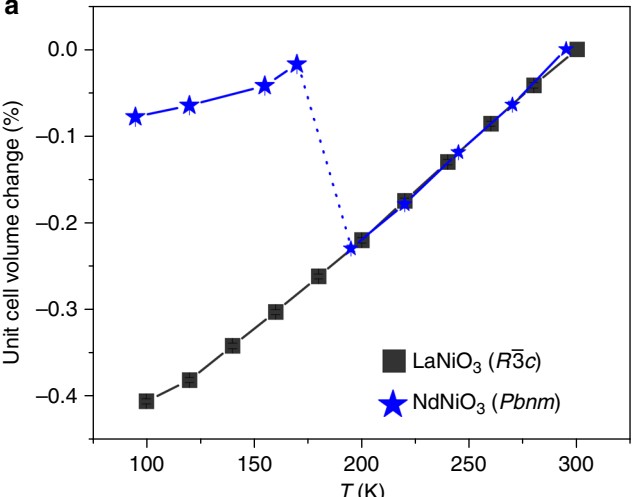

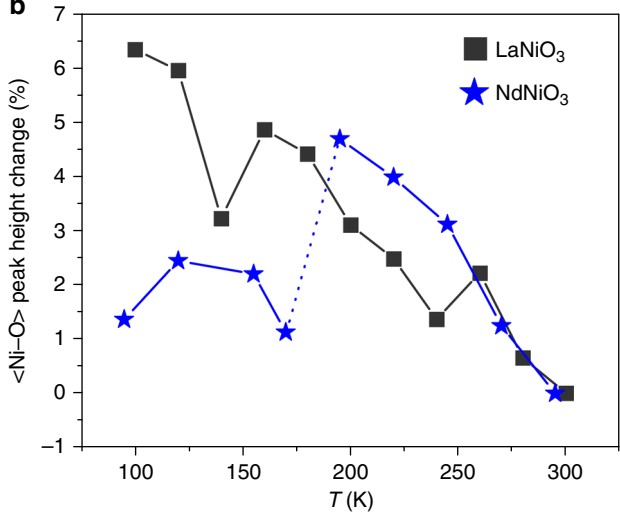

**Fig. 2** Temperature dependence of the unit cell parameters. **a** The change in unit cell volume for NdNiO$_3$ (blue stars) and LaNiO$_3$ (black squares) determined by neutron diffraction as a function of temperature $T$. For continuity, NdNiO$_3$ was refined using the orthorhombic (*Pbnm*) polymorph for the entire temperature range while LaNiO$_3$ was refined using the rhombohedral ($R\bar{3}c$) polymorph. **b** The peak height of the nearest-neighbor Ni–O correlation in the PDF of NdNiO$_3$ (blue stars) and LaNiO$_3$ (black squares). The error bars in both panels are smaller than the size of the data markers and were determined from uncertainty in the unit cell volume as determined through Rietveld refinement in GSAS, which were then normalized to the unit cell volume at room temperature

distance using Eq. (8). While this value is not straight forward to determine, we can obtain a rough estimate by using the Ni–Ni distance in a back of the envelope calculation. This assumption gives a mobility ~0.05 cm$^2$/V s, which is an order of magnitude smaller than the Hall mobility reported in ref. [30] for SmNiO$_3$ thin films. This value is not unphysically small, and may be due to differences in the samples (e.g., Sm vs. Nd/La or powders vs. films).

The behavior of $\sigma_{dc}$ and $\tau_e$ is in stark contrast to the Drude picture of metallic conduction, where conductivity decreases as $T$ increases due to thermally excited lattice vibrations and the carrier mobility subsequently decreases. Our data show that increasing $T$ instead enhances the carrier mobility in the nickelate metallic phase. This puzzling observation can be understood if the carriers are lattice polarons, as implied by the PDF data in Fig. 1.

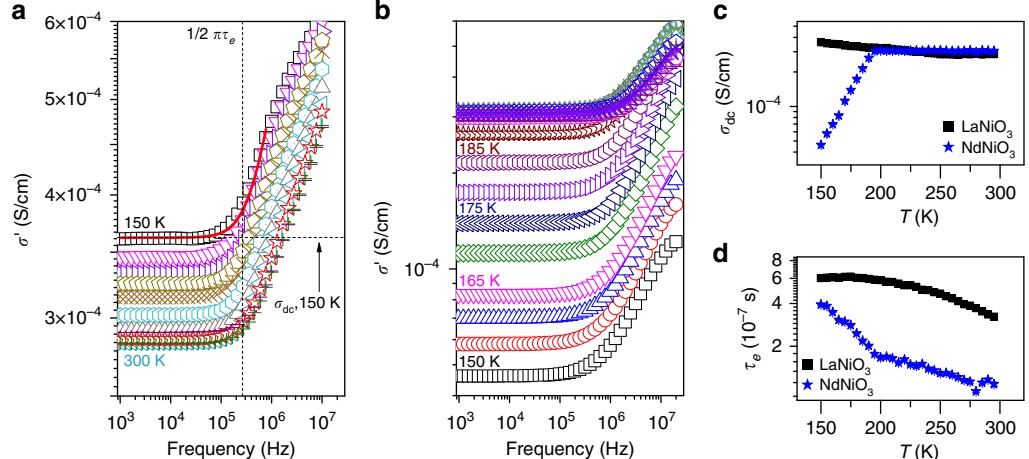

**Fig. 3** Broadband dielectric spectroscopy measurements. **a** The frequency dependence of the real part of complex conductivity ($\sigma'$) for LaNiO$_3$ from 300 to 150 K. The continuous-time-random walk (CTRW) model was fit to the spectra at all temperatures (shown as a solid red line for 150 K) to obtain values for dc conductivity (horizontal dashed line) and hopping time, $\tau_e$ (vertical dashed line). **b** The frequency dependence of the real part of $\sigma'$ for NdNiO$_3$ from 300 to 150 K. **c** Bulk dc conductivity of NdNiO$_3$ (blue stars) and LaNiO$_3$ (black squares) as a function of inverse temperature determined by fitting the CTRW model to broadband dielectric spectra. **d** Polaronic hopping time of NdNiO$_3$ (blue stars) and LaNiO$_3$ as a function of inverse temperature determined by fitting the CTRW model to broadband dielectric spectra

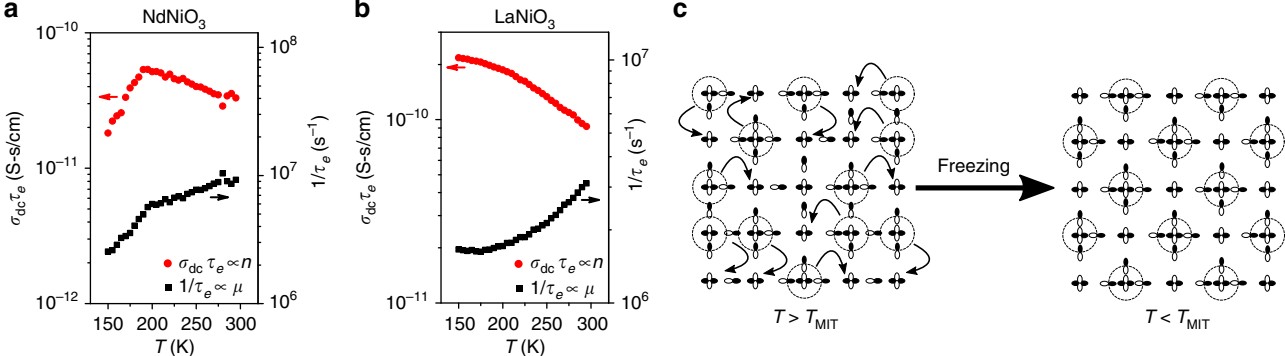

**Fig. 4** A sketch of the proposed bipolaronic condensation mechanism. **a** The dc conductivity ($\sigma'_{dc}$) multiplied by the carrier hopping time ($\tau_e$) (red circles, left axis) as well as the inverse hopping time (black squares, right axis) for NdNiO$_3$ as a function of temperature. The latter is proportional to the product of the carrier concentration ($n$) while the former is proportional to mobility ($\mu$). **b** The same for LaNiO$_3$. **c** A schematic sketch (projected in two dimensions) of the metal–insulator transition in NdNiO$_3$ in which a disordered liquid of polaronic distortions in the metallic phase freezes into a charge-ordered phase of lattice bipolarons. Circles denote double ligand holes, i.e., $d^8\underline{L}^2$, where two holes occupy the molecular orbitals on the surrounding O$_6$ octahedra. Here, we only show the $3d_{x^2-y^2}$ Ni orbital and surrounding O $2p_\sigma$ orbitals for simplicity

In this case, thermal excitations of the lattice make it easier to transfer the phonon cloud between neighboring sites and thus facilitate hopping.

The fact that the mobility in the metallic nickelates is enhanced with increasing $T$ means that carrier concentration must change with temperature to explain the apparent trends in conductivity. This is most evident when examining a plot of $\sigma_{dc}\tau_e$ (proportional to $n$) or $\tau_e^{-1}$ (proportional to $\mu_e$) vs. $T$ (Fig. 4). Conventional thought is that metals have a constant $n$, and the decrease in conductivity with temperature is caused instead by decreased mobility. We have shown that in both metallic NdNiO$_3$ (Fig. 4a) as well as LaNiO$_3$ (Fig. 4b), $\mu_e$ continually increases with temperature and it is instead $n$ that is decreasing. This result is in agreement with a recent optical spectroscopy study by Jaramillo et al.[31], which suggested that changes in carrier concentration are the origin of the bad metal behavior in the nickelates. This result can be understood in the framework of polaronic charge carriers proposed here, where the coherence between the phonon cloud and the holes becomes damped with increased temperature.

Interestingly, although $\mu_e$ does display a change of slope, the most profound feature of the MIT is the thermal behavior of $n$, which shows a maximum at the MIT and rapidly falls off at lower temperatures (Fig. 4a). This observation can be explained if there exists a highly disordered "liquid" of polaronic charge carriers in the metallic state that "freeze" into a periodic array in the insulating state, thus resulting in the dramatic decrease in the concentration of mobile charge carriers (Fig. 4c). This interpretation is in full agreement with the PDF data shown earlier, which revealed a sudden broadening of the Ni–O bond peak that was concomitant with the MIT.

## Discussion

In light of our results, it is interesting to speculate as to why LaNiO$_3$ does not undergo a MIT, while the remaining nickelates do. We believe that this is due to differences in the Ni–O–Ni bond angle $\theta$ produced by the variation on the $R$ ion's radius. In LaNiO$_3$, which has the largest $R$ ion, this angle is greatest

($\theta \sim 165°$), while it becomes smaller as $R$ decreases in size. (For example, in NdNiO$_3$ it is $\theta \sim 157°$ while for the end member LuNiO$_3$ it is estimated to be $\theta \sim 132.5°$ [3].) Decreasing the bond angle has two important effects in the context of polarons. First, it narrows the electronic bandwidth thus enhancing polaronic effects. Second, it controls the strength of the linear e–ph coupling. In a perfectly cubic structure, with a 180° Ni–O–Ni bond angle, the first order coupling to the oxygen displacement should vanish by symmetry. Thus, decreasing the bond angle from 180° will increase overall strength of the linear e–ph coupling. When combined with the reduced bandwidth, this effect should enhance the formation and localization of small polarons. A combination of these two effects may explain why LaNiO$_3$ does not undergo an MIT; however, further theoretical modeling will be necessary to determine which effect, if any, is dominant.

In summary, we have presented detailed neutron and broadband dielectric spectroscopy measurements on the rare-earth nickelates LaNiO$_3$ and NdNiO$_3$. Our results reveal evidence for distorted NiO$_6$ octahedra in the metallic phase of both materials, with only a minor change in carrier dynamics across the MIT. Our results provide a way to view the metallic state of the nickelates that differs somewhat from the previously proposed bond-disproportionation scenario. These results demonstrate that the $d^8\underline{L}^2$ complexes are pre-formed in the metallic state, forming a disordered (bi)polaronic liquid, with a bad metallic state driven by thermally activated transport properties at high temperature. Across the MIT, the compressed $d^8\underline{L}^2$ bipolarons condense into a charge-ordered lattice, as sketched in Fig. 4c. This understanding of the MIT in the rare-earth nickelates likely has analogies in other systems of negative charge transfer materials such as the high-temperature superconducting bismuthates and related systems [32,33]. This picture also provides natural explanations for both the observation of an isotope effect for $T_{MIT}$ [14] and Curie–Weiss behavior in non-magnetic LaNiO$_3$ [34]. In the latter case, the Ni 3$d^8$ ions in the expanded O$_6$ octahedra act as free $S = 1$ ions in the metallic phase.

## Methods

**Neutron total scattering experiments.** Polycrystalline LaNiO$_3$ and NdNiO$_3$ polycrystalline samples were prepared from citrate precursors obtained by a soft chemistry procedure. Stoichiometric amounts of analytical grade La$_2$O$_3$ or Nd$_2$O$_3$ and Ni(NO$_3$)$_2$·6H$_2$O were dissolved in a saturated acid citric solution containing some droplets of HNO$_3$. The citrate solution was slowly evaporated, leading to an organic resin that was dried at 120 °C and slowly decomposed at temperatures up to 600 °C. The products underwent a subsequent treatment in air at 800 °C for 2 h to eliminate all the organic materials and nitrates. The black precursor powders were heated under 200 bar of oxygen pressure for 12 h at 900 °C. Then, the samples were slowly cooled down to room temperature.

Neutron total scattering experiments were performed at the nanoscale ordered materials diffractometer (NOMAD) beamline at the Spallation Neutron Source at Oak Ridge National Laboratory [35]. Polycrystalline LaNiO$_3$ and NdNiO$_3$ samples were loaded into 2 mm diameter quartz capillaries to a height of ~1 cm. The samples were exposed to the neutron beam for ~1 h per measurement. An identical empty capillary was also measured for 1 h to correct for the background contribution to the data. LaNiO$_3$ was measured at 11 temperatures from 300 to 100 K, while NdNiO$_3$ was measured at nine temperatures from 295 to 95 K.

The total scattering structure function $S(Q)−1$ was obtained by normalizing scattering intensity from the sample (after the background was subtracted using the empty quartz capillary measurement) to that of a vanadium standard. The reduced PDF was calculated using the Fourier transform of $S(Q)−1$

$$G(r) = \frac{2}{\pi} \int_{Q_{min}}^{Q_{max}} Q[S(Q)-1]\sin(Qr)dQ, \quad (1)$$

where $Q$ is the scattering vector and is defined as $Q = 4\pi/\lambda \sin\theta$. Here, $\lambda$ and $\theta$ are the neutron wavelength and scattering angle, respectively. A value of 0.1 and 31.4 Å$^{-1}$ was used for $Q_{min}$ and $Q_{max}$, respectively, in the Fourier transform. The measured $S(Q)−1$ was multiplied by a Lorch function [36] before the transformation to exclude the effects of artificial ripples on data analysis.

Unit cell volumes were determined using Rietveld refinement of the highest Q-resolution backscattering bank of diffraction data with GSAS software [37]. PDFgui [38] was used for real-space refinement of PDF data, while PDF peak heights were

determined using peak fitting in OriginPro with a Gaussian function after subtracting a linear baseline.

**Broadband dielectric spectroscopy measurements.** Broadband dielectric spectroscopy measurements were performed using a Novocontrol Alpha Analyzer with a frequency range of 10$^3$–10$^7$ Hz. The temperature was varied from 300 to 150 K using a QUATRO liquid nitrogen controller with a stability limit of ±0.1 K. Powder samples were hot pressed at 150 °C into aluminum foil to a thickness of 50 μm. The aluminum foil served as the bottom electrode while a 6 mm diameter brass electrode was used for the top. Teflon spacers were used to create a gap of 100 μm between the sample and top electrode. Complex conductivity ($\sigma' + i\sigma''$) was calculated from the measured impedance data using WinDeta software from Novocontrol.

**Data availability.** The data that support the findings of this study are available from the corresponding author upon reasonable request.

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

## Acknowledgements

The authors thank M. Berciu, N. C. Plumb, G. Sawatzky, and M. Fitzsimmons for stimulating discussions. This work was supported by University of Tennessee's Office of Research & Engagements Organized Research Unit program. A portion of this research used resources at the Spallation Neutron Source, a DOE Office of Science User Facility operated by the Oak Ridge National Laboratory. H.Z. acknowledges support from NSF-DMR-1350002. J.A.A. acknowledges the Spanish MINECO for granting the project MAT2013-41099-R. Collection and interpretation of PDF was supported as part of the Materials Science of Actinides, an Energy Frontier Research Center funded by the US Department of Energy, Office of Science, Basic Energy Sciences under Award # DE-SC0001089.

## Author contributions

J. Shamblin led the experimental effort and analyzed all of the data. J. Shamblin, J.N., and M.L. performed the neutron scattering experiments. J. Shamblin, M.H., and J. Sangoro performed the broadband dielectric spectroscopy experiments. J.A.A. and H.Z. were responsible for sample preparation and characterization. S.J. conceived of the project and was responsible for project management. J. Shamblin and S.J. wrote the paper with input from all authors.

## Additional information

**Competing interests:** The authors declare no competing financial interests.

