## [Peer Review File · Nature Communications]

Reviewers' comments:

Reviewer #1 (Remarks to the Author):

The work by Shamblin et al utilizes total neutron scattering experiments to study the local structure of the nickelates LaNiO_3 and NdNiO_3 , and thereby try to explain how the structure reveals the nature of the metal-to-insulator in the Nd-system. Dielectric broadband spectroscopy is also utilized to characterize carrier dynamics in both compounds. The driving point of the manuscript is that through analysis of pair distribution functions (PDFs) of the nickelate phases from neutron scattering, these materials should be 'bad metals' at all temperatures and have a polaronic mechanism for conductivity. The only difference is that for the Nd-case, these polarons form long-range order below 200 K due to carrier concentration effects.

The manuscript presents high quality neutron data, and the analysis itself is of high quality. I do not doubt the interpretation of the neutron PDFs, either, it seems very likely that the monoclinic symmetry is the best description of the local coordination for both of the nickelates. The study does merit publication in Nature Communications given the growing interest in quantum materials such as these nickelates, where quasiparticles dominate the macroscopic properties. I do have some issues, however, with the presentation of this work. Namely, the main text needs some more work, so that the impact in the field of quantum materials is broader. There were some unanswered questions as well, that may in fact be more tied to presenting the overall conclusions more clearly. I list some of these questions and issues below:

- 1.) If polarons are present in both LaNiO_3 and NdNiO_3 and they affect the carrier concentration, then why does the La compound not undergo the MIT at 200 K as in the NdNiO_3 ? This was not clear to me. I get that one displays freezing of these polarons, and the other doesn't from the PDFs. But why? I am not sure this very important question was asked. After all La and Nd are isovalent and therefore there should only be $d^{7/8}$ in both compounds.
- 2.) How does the polaron model ($d^{8/L2}$) from Figure 3c correspond to the local structural distortions observed in the PDF? I get that it leads to a lowering of symmetry, but maybe the point needed to be made more strongly that the two Ni sites of the monoclinic structure correspond to the two different Ni states of the polaron model (above and below the MIT). Or is my interpretation an oversimplification of the model? I thought the manuscript could have driven this point more consistently and boldly in the text.
- 3.) How does a $d^{7/8}$ octahedral configuration lead to a single electron in the t_{2g} manifold? Did the authors mean a single hole in the t_{2g} manifold? The authors make this claim on page 2, paragraph 3.
- 4.) When the authors state that the formula unit of NiO_3 is best described as $\alpha|d^{7/8} + \beta|d^{8/L}$, I was a bit confused. What do they mean by formula unit? Do they mean the wavefunction describing the electronic configuration? They may be familiar with this language, but for Nature Communications they will need to write with less jargon and more descriptive terms.
- 5.) The last sentence of paragraph 1 of page 3 is confusing and grammatically incorrect (there are two verbs here but only one sentence). Also, why do the two holes localize on one oxygen for one Ni atom?
- 6.) The authors mention bond-disproportionation once, not again in the manuscript. Does this imply that charge-disproportionation is the correct mechanism but not bond-disproportionation?

7.) The authors may want to explain what a small-box refinement is. Since they did not perform a large-box refinement, I did not even see the point of describing it as such.

8.) What does it mean for differences between the refinements to be tiny? Seems too qualitative of a description.

9.) Perhaps Fig. 1 would benefit by either having the Rwp's or the difference curves. It is not entirely obvious from the figure that certain models are superior to the others.

10.) Do the symbols of Fig. 2 have error bars? And if so, are they smaller than the symbol size? This is particularly important for Fig 2b, where one is trying to see if these curves show statistically important trends.

Reviewer #2 (Remarks to the Author):

This paper reports on the mechanism for the metal-insulator transition (MIT) in rare-earth nickelates. Recently it gains a broad consensus that rare-earth nickelates have negative charge transfer and the MIT is driven by the formation of the long-range ordering of holes at oxygen sites. This paper reveals, from neutron scattering and broadband dielectric spectroscopy, that such a disproportionation of oxygen holes remains even in the metallic phase as a disordered polaronic state. The transport properties in the metallic phase are governed by the dynamic fluctuation of the polarons.

The above picture of the MIT is in line with the conclusion of Ref. 3. The present paper shows more direct evidences for this picture. The reviewer considers this paper gives a deeper insight into the MIT in nickelates and will promote theoretical challenge to fully describe the complex feature of the MIT. Therefore, this paper is worth publishing in Nature Communications if the authors can respond to all the following questions.

1. According to the previous paper by one of the authors (Ref. 13), the metallic state of NdNiO₃ is uniformly self-doped Mott insulating state (d₈L) and the MIT occurs associated with the change in the electronic structure from (d₈L)(d₈L) to (d₈L₂)-(d₈). How does the authors explain the inconsistency between the previous and present papers ?

2. Judging from the PDF results, the metallic state of LaNiO₃ seems to have larger local monoclinic distortion compared to NdNiO₃. Why does LaNiO₃ show stronger tendency toward the bond disproportionation ?

3. The peak height of the nearest neighbor Ni-O correlation in NdNiO₃ decreases at the transition temperature. Why is the peak broader in the long-range ordered phase?

4. Why does NdNiO₃ have shorter τ_e (or equally, larger mobility) compared to LaNiO₃? Is it consistent with the result of Hall effect ? Considering the band width, the reviewer expect opposite relation.

5. Is the temperature dependences of n and μ estimated from τ_e consistent with those obtained from other measurement techniques such as Hall effect ?

6. It is true that the air-gap structure does not affect the frequency dependence of the AC conduction considering the equivalent circuit depicted in Fig. S5. However, the reviewer feels that the measurement of AC conduction using such an air-gap structure is not so established way. Have the authors confirmed that AC conduction measurements with similar structure for well-known

materials gives reasonable value and temperature dependence of τ_e ?

Minor comments:

1. Page 2, line 7 driven many-body -> driven by many-body ?
2. Page 2, line 8 attracted attracted -> One "attracted" is needed to be removed.
3. Page 5, line 5 Ref. 26 -> Reference number is described by superscript.

Reviewer #3 (Remarks to the Author):

In this manuscript 'Experimental evidence for bipolaron condensation as a mechanism for the metal-insulator transition in rare-earth nickelates', the author Shamblin et al. reported pair distribution, and broadband dielectric spectroscopy measurement of polycrystalline LaNiO₃ and NdNiO₃ to understand the metal insulator transition of rare-earth nickelate family.

Bulk LaNiO₃ is rhombohedral and NdNiO₃ undergoes a metal-insulator transition with a simultaneous lowering of symmetry from orthorhombic to monoclinic. The authors have concluded from the fitting of pair distribution function that the monoclinic model fits the data better over the entire range of temperature for NdNiO₃ (both metallic and insulating phase) and LaNiO₃. The presence of short range monoclinic/'bond disproportionation' phase in metallic phase had been reported earlier using EXAFS (Piamonteze et al. Phys. Rev. B 71, 012104 (2005)) and by musr (Physica B 374-375, 87-90 (2006)). The authors in this present manuscript have similar conclusion with a different experimental technique. However, the description of metal-insulator transition in polaron picture using the results of structural analysis and broad band spectroscopy is new and interesting.

I will be happy to recommend the manuscript for publication in Nature Communication if the authors can satisfactorily explain the followings:

1. Can the author explain how one can get long range orthorhombic or rhombohedral structure from a collection of short range monoclinic structures?
2. The conclusions, obtained from the broad band dielectric spectroscopic study depend on the fitting of real part of ac conductivity by a continuous-time random work model. As the hopping charge carriers are facing randomly varying energy barriers in this model, the author should describe why this model is suitable for the present cases. What is the source of random barriers?
3. How does the length of airgap affect the results of dielectric spectroscopy? The results in the manuscript are for a 100 micrometer airgap. As the airgap size of experiments should not affect the intrinsic time scale of the samples, the author should provide a complete set of results and analysis with another air gap size.
4. Do the authors also observe hysteresis in these measurements?

Other comments:

The author should cite previous papers on short range monoclinic distortion in metallic phase of nickelates.

Response to Reviewer One

The reviewer wrote: *The work by Shamblin et al utilizes total neutron scattering experiments to study the local structure of the nickelates LaNiO_3 and NdNiO_3 , and thereby try to explain how the structure reveals the nature of the metal-to-insulator in the Nd-system. Dielectric broadband spectroscopy is also utilized to characterize carrier dynamics in both compounds. The driving point of the manuscript is that through analysis of pair distribution functions (PDFs) of the nickelate phases from neutron scattering, these materials should be 'bad metals' at all temperatures and have a polaronic mechanism for conductivity. The only difference is that for the Nd-case, these polarons form long-range order below 200 K due to carrier concentration effects.*

The manuscript presents high quality neutron data, and the analysis itself is of high quality. I do not doubt the interpretation of the neutron PDFs, either, it seems very likely that the monoclinic symmetry is the best description of the local coordination for both of the nickelates. The study does merit publication in Nature Communications given the growing interest in quantum materials such as these nickelates, where quasiparticles dominate the macroscopic properties. I do have some issues, however, with the presentation of this work. Namely, the main text needs some more work, so that the impact in the field of quantum materials is broader. There were some unanswered questions as well, that may in fact be more tied to presenting the overall conclusions more clearly. I list some of these questions and issues below:

Our response: We thank the referee for their time and effort in reviewing our work. We appreciated their obvious interest in our work and their detailed comments. After reviewing the comments from the referees, we also agree that some of the main ideas could have been presented in a better fashion throughout the paper. We have attempted to do so in the revised version.

Before we launch into our response, we would like to apologize to the referee for the long delay in resubmitting the work. Motivated by the referee reports, we decided to carry out additional measurements for the dielectric response of our samples but discovered that our samples had degraded since we conducted our initial experiments. We, therefore, had to synthesis new samples before carrying out the new measurements, which took some time. Because of this, we are now able to provide both concrete responses to the relevant referee comments and a second completely independent data set (obtained by a different student and on different samples). Since this extended data set speaks to the reproducibility of our results, we have also included it in the supplementary text of the paper.

The reviewer wrote: *1.) If polarons are present in both LaNiO_3 and NdNiO_3 and they affect the carrier concentration, then why does the La compound not undergo the MIT at 200 K as in the NdNiO_3 ? This was not clear to me. I get that one displays freezing of these polarons, and the other doesn't from the PDFs. But why? I am not sure this very important question was asked. Afterall La and Nd are isovalent and therefore there should only be Ni d^7 in both compounds.*

Our response: We agree that both La and Nd will not introduce additional carriers into the Ni and O derived bands crossing the Fermi level and so the NiO_3 units should have similar electronic states; however, we disagree that they should be regarded as d^7 . As argued in several prior theoretical and experimental studies, we believe that the correct starting point is the $|d^8\bar{\underline{L}}\rangle$ electronic configuration, where one of the would-be holes on the Ni site has transferred to the surrounding oxygen (denoted $\bar{\underline{L}}$). This charge configuration is important because it is known to couple strongly to the Ni-O bond-stretching phonon modes, as discussed in Park *et al.* [PRL **109**, 156402 (2012)] and Johnston *et al.*, [PRL **112**, 106404 (2014)], and has experimental support from a recent RIXS experiment [Nat. Commun. **7**, 13017 (2016)] and many prior core-level spectroscopies. Our current work expands on this idea and demonstrates that the e-ph coupling to the bond-stretching modes is active in the metallic phase, leading to polaronic-like charge carriers. We propose that it is the condensation of these carriers into an ordered state that occurs across the MIT.

Why LaNiO_3 remains metallic while NdNiO_3 undergoes an MIT is an important question. We hypothesize that this is due to the Ni-O-Ni bond angles, which are different for the two systems; in LaNiO_3 the bond angle is $\sim 165^\circ$, while in NdNiO_3 it is $\sim 157^\circ$ [M. L. Medarde, J. Phys. Condens. Matter **9**, 1679 (1997)]. This difference has two important effects on polaronic carriers. The **first** is that it controls the electronic bandwidth of the material and the tendency towards polaronic effects should be larger as the bandwidth decreases. **Second**, the bond angle also controls the strength of the linear e-ph coupling. Here, the breathing distortions couple to the holes on the oxygen sites through the modulation of the Ni-O hopping integrals. In a perfectly cubic structure, with a 180° angle, the first order coupling to the oxygen displacement should vanish by symmetry and deviations from this value introduce a linear coupling. The increasing deviation from 180° in going from La to Pr to Nd and so forth will therefore increase the total coupling. When combined with the reduced bandwidth, this should enhance the tendency to form and localize small polarons. We believe that it is a combination of these two effects that accounts for the difference between La and Nd, as well as the systematic increase in T_{MIT} as the radius of the rare earth ion decreases further. We are currently carrying out more detailed QMC calculations to demonstrate this more explicitly, which will be published at a later date.

We agree that these points should be discussed in the manuscript, and we have added a paragraph to the discussion section that does so.

The reviewer wrote: 2.) *How does the polaron model (d^8L^2) from Figure 3c correspond to the local structural distortions observed in the PDF? I get that it leads to a lowering of symmetry, but maybe the point needed to be made more strongly that the two Ni sites of the monoclinic structure correspond to the two different Ni states of the polaron model (above and below the MIT). Or is my interpretation an oversimplification of the model? I thought the manuscript could have driven this point more consistently and boldly in the text.*

Our response: Your interpretation is in line with ours and this is what we tried to convey. Our PDF data indicates that both contracted and expanded NiO_6 octahedra are present in **both** the metallic and insulating phases of the LaNiO_3 and NdNiO_3 samples. The two inequivalent Ni sites in the monoclinic model correspond to these two types of local environments. We also found that the metallic state data for both samples could be well fit on long length scales using a cubic structure. This indicates that the contracted and expanded NiO_6 appear uniform when viewed on longer length scales. These results can be reconciled if the contracted NiO_6 are disordered throughout the material, suggesting the proposed polaron picture. Of course they could be statically disordered, but our broadband dielectric spectroscopy results indicated that this is unlikely. We have modified the discussion following the PDF results to make this point clearer to the reader.

The reviewer wrote: 3.) *How does a d^7 octahedral configuration lead to a single electron in the t_{2g} manifold? Did the authors mean a single hole in the t_{2g} manifold? The authors make this claim on page 2, paragraph 3.*

Our response: This is an unfortunate typo. We meant to say that the d^7 configuration would place a single electron in the e_g manifold. We thank the referee for bringing this to our attention.

The reviewer wrote: 4.) *When the authors state that the formula unit of NiO_3 is best described as $\alpha d^7 + \beta l d^8 L$, I was a bit confused. What do they mean by formula unit? Do they mean the wavefunction describing the electronic configuration? They may be familiar with this language, but for Nature Communications they will need to write with less jargon and more descriptive terms.*

Our response: We did in fact mean the electronic wavefunction. Here we were using notation that was familiar to us from prior spectroscopy work. For a correlated material it is common to start from an atomic picture and then add electron itinerancy to describe the electronic wavefunction. For example,

when one says that the Ni is in a d^7 configuration it means that if orbital overlaps were zero then the Ni would be a d^7 state. The $d^8\bar{L}$ state means that without orbital overlaps, one would have two holes on the Ni and one hole occupying a molecular orbital formed from the ligand O atoms surrounding the Ni. After orbital overlaps are introduced, the d^7 and $d^8\bar{L}$ states mix, resulting in a wavefunction of the form $\alpha|d^7\rangle + \beta|d^8\bar{L}\rangle$. The fact that $\beta > \alpha$ means that the $d^8\bar{L}$ atomic state is lower in energy than the d^7 . We agree that this notation is probably not so familiar for the broader readership, so we have revised it in the introduction to better explain this view.

The reviewer wrote: 5.) *The last sentence of paragraph 1 of page 3 is confusing and grammatically incorrect (there are two verbs here but only one sentence). Also, why do the two holes localize on one oxygen for one Ni atom?*

Our response: We thank the reviewer for bringing this to our attention. We have revised this sentence to now read “The insulating phase corresponds to a crystal structure where the oxygen sublattice has contracted around alternating Ni sites along the three cubic crystallographic axes. This structure results in two inequivalent Ni sites and a monoclinic phase, consistent with experiments. The Ni site surrounded by the compressed oxygen octahedron has a ($d^8\bar{L}^2$) charge configuration, where two holes occupy a molecular orbital formed from the ligand oxygen orbitals with e_g symmetry. The Ni at the center of the expanded octahedron has a ($3d^8$) configuration.”

The reviewer wrote: 6.) *The authors mention bond-disproportionation once, not again in the manuscript. Does this imply that charge-disproportionation is the correct mechanism but not bond-disproportionation?*

Our response: We believe that neither the charge-disproportionation nor bond-disproportionation description provides a complete and correct description of this transition; however, the bond-disproportionation scenario is closer to what happens in nature. In the charge-disproportionation scenario, the Ni undergo a change in valence from (d^7)(d^7) to ($d^{7+\delta}$)($d^{7-\delta}$), where the charge is predominantly on the Ni sites. In this scenario, the Ni-O bonds lengths distort in response to the change in valence. In the bond-disproportionation scenario, the charges in the metallic phase are viewed as ($d^8\bar{L}$) where one has one hole per three oxygen atoms. In this scenario, the Ni-O bond lengths change coherently throughout across T_{MIT} , resulting in a reorganization of the wavefunction such that ($d^8\bar{L}$)($d^8\bar{L}$) \rightarrow ($d^8\bar{L}^2$)(d^8). Our results show that the distorted octahedra are preformed in the metallic state and are ordering across T_{MIT} . The primary difference is that the metallic phase should be viewed as a disordered gas of ($d^8\bar{L}^2$), ($d^8\bar{L}$), and (d^8) configurations. We choose not to call this bond-disproportionation to emphasize this difference. However, since many readers use this language, we have revised some text to address this issue.

The reviewer wrote: 7.) *The authors may want to explain what a small-box refinement is. Since they did not perform a large-box refinement, I did not even see the point of describing it as such.*

Our response: We believe this is a fair comment, but we felt it important to interested neutron scattering experts to explicitly say what method we used for modeling the data. We would prefer to leave it as is unless the reviewer has a strong objection.

The reviewer wrote: 8.) *What does it mean for differences between the refinements to be tiny? Seems too qualitative of a description.*

Our response: We agree that this description was poorly worded and have updated our discussion as well as Figure 1 to include goodness-of-fit values. Qualitatively, the fits of the orthorhombic and monoclinic structures are extremely similar across the entire fitting range except for a subtle difference in the position

of the first peak corresponding to the NiO₆ octahedra, which is slightly displaced using the orthorhombic model.

The reviewer wrote: 9.) *Perhaps Fig. 1 would benefit by either having the Rwp's or the difference curves. It is not entirely obvious from the figure that certain models are superior to the others.*

Our response: We agree with the referee and have added Rwp values to each curve.

The reviewer wrote: 10.) *Do the symbols of Fig. 2 have error bars? And if so, are they smaller than the symbol size? This is particularly important for Fig 2b, where one is trying to see if these curves show statistically important trends.*

Our response: The symbols in both panels of Fig. 2 have errors bars, but they are smaller than the symbols. We have added a statement to this effect in the figure caption. We believe that the two points in Fig. 2b that deviate from the trend of increasing peak height with decreasing temperature for LaNiO₃ are due to artifacts arising from the Fourier transform and are not of physical origin. This believe is further evidenced by taking into account the O-O peak height in Supplemental Fig. 3, where the correlations associated with neighboring NiO₆ octahedra show a clear broadening across the *MIT* for NdNiO₃ that is not present in LaNiO₃.

Response to Reviewer Two

The Reviewer wrote:

This paper reports on the mechanism for the metal-insulator transition (MIT) in rare-earth nickelates. Recently it gains a broad consensus that rare-earth nickelates have negative charge transfer and the MIT is driven by the formation of the long-range ordering of holes at oxygen sites. This paper reveals, from neutron scattering and broadband dielectric spectroscopy, that such a disproportionation of oxygen holes remains even in the metallic phase as a disordered polaronic state. The transport properties in the metallic phase are governed by the dynamic fluctuation of the polarons.

The above picture of the MIT is in line with the conclusion of Ref. 3. The present paper shows more direct evidences for this picture. The reviewer considers this paper gives a deeper insight into the MIT in nickelates and will promote theoretical challenge to fully describe the complex feature of the MIT. Therefore, this paper is worth publishing in Nature Communications if the authors can respond to all the following questions.

Our Response: We first would like to thank the reviewer for their time and interest in our work. We are also encouraged by their positive appraisal of our work. Before we launch into our response, we would like to apologize for the long delay in resubmitting the paper. Motivated by the original referee reports, we decided to carry out additional measurements for the dielectric response of our samples but discovered that our samples had degraded in storage. We, therefore, had to synthesis new samples before carrying out the new measurements, which took some time. Because of this, we are now able to provide both concrete responses to the relevant referee comments and a second completely independent data set (obtained by a different student and on different samples). Since this extended data set speaks to the reproducibility of our results, we have also included it in the supplementary text of the paper.

The reviewer wrote: *1. According to the previous paper by one of the authors (Ref. 13), the metallic state of NdNiO₃ is uniformly self-doped Mott insulating state (d^8L) and the MIT occurs associated with the change in the electronic structure from (d^8L)(d^8L) to (d^8L_2)-(d^8). How does the authors explain the inconsistency between the previous and present papers?*

Our Response: We do not believe there is an inconsistency between the two results but rather a revision of our thinking of the metallic state in light of our new results. In Ref. (13) we examined the reorganization of the electronic structure and wavefunction and that would occur if one began from the ($d^8\bar{L}$) viewpoint and introduced coupling to the *collective* breathing distortion of the lattice. There, we were trying to determine if the MIT could be accounted for if one viewed the lattice displacement as the causal agent rather than as a symptom of charge disproportionation. The main point of that work was that the bond disproportionation could still occur without transferring net charge between the Ni sites. We would also like to stress that in that paper we considered the collective breathing motion of the lattice, where each O octahedra was expanded and contracted along the $\mathbf{Q} = (\pi, \pi, \pi)/a$ direction; the model did not allow for breathing distortions to occur around randomly distributed Ni ions. The theory model was therefore unable to capture the polaronic state we are advocating here.

When we wrote Ref. (13) we had speculated internally that the e-ph interaction was active in the metallic state as we claimed here, and that should lead to some collection of “pre-distorted” octahedra. (This speculation is what motivated the current study.) But in the absence of PDF data, we had no evidence to support this view that so we took the simpler CDW-formation view. However, if the e-ph coupling were present in the metallic phase, even ($d^8\bar{L}$) configurations should have some degree of lattice distortion surrounding them. In this case, it is possible that the distribution of Ni-O bonds comes from a fluctuation collection of (d^8), ($d^8\bar{L}$), and ($d^8\bar{L}^2$), configurations, which then freeze into the ordered phase at the MIT. This scenario would be consistent with our data, as well as the bond disproportionation without

charge transfer picture developed previously. We have added some additional discussion to the main text of the paper to address these points.

The reviewer wrote: 2. *Judging from the PDF results, the metallic state of LaNiO₃ seems to have larger local monoclinic distortion compared to NdNiO₃. Why does LaNiO₃ show stronger tendency toward the bond disproportionation ?*

Our Response: We appreciate the reviewer for bringing this up as this may serve as a source of confusion. The conclusion that LaNiO₃ shows a larger local monoclinic distortion than NdNiO₃ is incorrect. Even though the monoclinic structure resulted in the best PDF refinement for both samples, the PDFs remain slightly different for both samples. For example, at 300 K, the disproportionation in the volume of the contracted and expanded NiO₆ octahedra (as determined by the refinements shown in Fig. 1) is larger for NdNiO₃ (9.51 Å³ and 10.10 Å³) than for LaNiO₃ (9.73 Å³ and 10.02 Å³).

The reviewer wrote 3. *The peak height of the nearest neighbor Ni-O correlation in NdNiO₃ decreases at the transition temperature. Why is the peak broader in the long-range ordered phase?*

Our Response: This observation is one of the key observations in our manuscript. Due to thermal motion, the resolution of any experimentally attainable PDF is insufficient to de-convolve into the contributions from the contracted and expanded NiO₆ octahedra. The degree of distortion can instead be determined by the peak height (or equally width) of the nearest-neighbor Ni-O correlation. In the metallic phase, when these distortions are smaller and/or fluctuating and mobile, the Ni-O correlations of the contracted and expanded octahedra are more “smeared” and thus centered over the average Ni-O distance. At the MIT, the contracted and expanded octahedra freeze into place resulting in a narrower peak due to the overlap of the more localized shorter and longer Ni-O correlations.

The Referee Wrote: 4. *Why does NdNiO₃ have shorter τ_e (or equally, larger mobility) compared to LaNiO₃? Is it consistent with the result of Hall effect ? Considering the band width, the reviewer expect opposite relation.*

Our Response: One should not compare the τ_e extracted from the broadband dielectric spectroscopy (BDS) data to the Drude model τ obtained in a Hall or Drude measurement; the BDS τ_e is a hopping time for the carriers and not a scattering rate. Our results indicate that hopping time is shorter in NdNiO₃, which may be related to the samples themselves, or to the difference in the Ni-O-Ni bond angle, which is smaller in NdNiO₃ (Please refer to our response to referee 1 and our revised discussion on what this difference does to the polaron). We have performed a second set of measurements on a new set of La and Nd samples, and found that τ_e has some sample dependence (see the modified supplementary materials). We attribute this to sample quality as small polarons can be strongly pinned by defects in the sample (see for example, Ebrahimnejad and Berciu PRB **88**, 104410).

The Referee Wrote: 5. *Is the temperature dependences of n and μ estimated from τ_e consistent with those obtained from other measurement techniques such as Hall effect ?*

6. *It is true that the air-gap structure does not affect the frequency dependence of the AC conduction considering the equivalent circuit depicted in Fig. S5. However, the reviewer feels that the measurement of AC conduction using such an air-gap structure is not so established way. Have the authors confirmed that AC conduction measurements with similar structure for well-known materials gives reasonable value and temperature dependence of τ_e ?*

Our response: Points 5 and 6 are related so we will address them both here. To our knowledge, the charge carrier hopping time, τ_e , has not been measured for such materials. This may not be technically

feasible as the hopping time will be on the order of the inverse bandwidth and therefore extremely short. The scattering rate ($1/\tau$) has been measured in numerous studies, but this τ is a fundamentally different value than reported in our manuscript. Converting our τ_c to carrier mobility, requires a knowledge of the jump distance per the equation on page 8 of our manuscript, which is not so straight forward to determine. Simply using the NiO_6 - NiO_6 spacing as an approximate jump distance in a back of the envelope calculation gives a mobility on the order of $0.05 \text{ cm}^2/\text{V.s}$, which is an order of magnitude smaller than the Hall mobility reported in Ha *et al.*, Phys. Rev. B **87**, 125150 (2013) but is not unphysically small. Again, this may be related to sample quality/differences; the values reported by Ha *et al.* are also for SmNiO_3 thin films rather than bulk NdNiO_3 and LaNiO_3 powders. We have inserted a line into the main text pointing out these differences and discussing our thoughts on their origin. We believe that this will stimulate further experimental work.

The Referee wrote:

Minor comments:

1. Page 2, line 7 *driven many-body* -> *driven by many-body* ?
2. Page 2, line 8 *attracted attracted* -> One “*attracted*” is needed to be removed.
3. Page 5, line 5 *Ref. 26* -> *Reference number is described by superscript.*

Our Response: Thank you for bringing these to our attention. We have corrected them in the revised version of the manuscript. The subscript reference is due to the latex template we are currently using, which does not seem to accept the `\onlinecite{}` command.

Response to Reviewer Three

The reviewer wrote: *In this manuscript 'Experimental evidence for bipolaron condensation as a mechanism for the metal-insulator transition in rare-earth nickelates', the author Shamblin et al. reported pair distribution, and broadband dielectric spectroscopy measurement of polycrystalline LaNiO₃ and NdNiO₃ to understand the metal insulator transition of rare-earth nickelate family.*

Bulk LaNiO₃ is rhombohedral and NdNiO₃ undergoes a metal-insulator transition with a simultaneous lowering of symmetry from orthorhombic to monoclinic. The authors have concluded from the fitting of pair distribution function that the monoclinic model fits the data better over the entire range of temperature for NdNiO₃ (both metallic and insulating phase) and LaNiO₃. The presence of short range monoclinic/'bond disproportionation' phase in metallic phase had been reported earlier using EXAFS (Piamonteze et al. Phys. Rev. B 71, 012104 (2005)) and by musr (Physica B 374-375, 87-90 (2006)). The authors in this present manuscript have similar conclusion with a different experimental technique. However, the description of metal-insulator transition in polaron picture using the results of structural analysis and broad band spectroscopy is new and interesting.

I will be happy to recommend the manuscript for publication in Nature Communication if the authors can satisfactorily explain the followings:

Our Response: We first thank the referee for their time and effort in reviewing our work and for his/her detailed comments. Before we launch into our response, we would like to apologize to the referee for the long delay in resubmitting the work. Motivated by the referee reports, we decided to carry out additional measurements for the dielectric response of our samples but discovered that our samples had degraded in storage. As a result, we had to synthesis new samples before carrying out the new measurements, which took some time. Because of this, however, we are now able to not only provide concrete responses to the relevant referee comments but also provide a second completely independent data set (obtained by a different student and on different samples). Since this extended data set speaks to the reproducibility of our results, we have also included it the revised supplementary materials.

The reviewer wrote:

1. Can the author explain how one can get long range orthorhombic or rhombohedral structure from a collection of short range monoclinic structures?

Our response: This question is important and addressing it cuts to the heart of our conclusions. Recent experiments have actually shown that this is perhaps not such an uncommon phenomenon (see for example, Li et al. [Advanced Electronic Materials 2, 2016], Shamblin et al. [Nature Materials 15, 2016] among others). As an oversimplification, from a local structure standpoint, the primary difference between the monoclinic and orthorhombic or rhombohedral structures, is the presence of collapsed and expanded NiO₆ octahedra. So long as these collapsed and expanded octahedra are randomly dispersed throughout the crystal matrix, the long-range structure will appear to be of a higher symmetry when averaged over the different randomly placed NiO₆ octahedra.

The reviewer wrote:

2. The conclusions, obtained from the broad band dielectric spectroscopic study depend on the fitting of real part of ac conductivity by a continuous-time random work model. As the hopping charge carriers are facing randomly varying energy barriers in this model, the author should describe why this model is suitable for the present cases. What is the source of random barriers?

Our response: We agree with the reviewer that this point should be phrased more clearly in the manuscript. Our data suggests that the charge carriers are polarons and bi-polarons in both the metallic and insulating states. In the metallic state, the PDF data suggests these form a disordered liquid or gas,

which serves as the source of random barriers; neighboring polarons will feel a different barrier to hopping depending on whether or not the neighboring Ni site also hosts a polaron or bipolaron. We have inserted new text into the paper to explain this to the reader.

The reviewer wrote:

3. How does the length of airgap affect the results of dielectric spectroscopy? The results in the manuscript are for a 100 micrometer airgap. As the airgap size of experiments should not affect the intrinsic time scale of the samples, the author should provide a complete set of results and analysis with another air gap size.

Our response: The air gap geometry is used often in the study of amorphous solids and polymers, where it does not affect the dynamics extracted from BDS measurements [see for example: Serghei *et al.*, J. Chem Phys. **131**, 154904 (2009), Tress *et al.*, Science **341**, 1371 (2013), and Heres *et al.*, ACS Macro Lett. **5**, 1065 (2016)]. However, since the air gap geometry is comparatively new for quantum materials, we agree that additional data would add confidence to our results. We have, therefore, carried out an additional set of measurements on a completely new set of samples (as indicated, our original samples had degraded in storage) but this time with a 50 μm air gap instead of the original 100 μm air gap used in the main text. The results are summarized below in Fig. R1, which also included in the revised supplementary materials.

Figure R1: The temperature dependence of the dc conductivity (left) and hopping time (right) for LaNiO₃ and NdNiO₃ samples with a 50 μm and 100 μm thick air gap. Note that the measurements were obtained on two distinct sets of samples (different powder pressings). These results also produce qualitatively similar temperature dependencies of the mobility and carrier concentration as a function of temperature (lower panels).

The measurements were performed on two different NdNiO₃ and LaNiO₃ samples. We believe that the magnitude of conductivity changes between the measurements because the compacted powders vary from sample to sample. However, it is clear that the temperature dependence of the dc conductivity is not sensitive to the air gap. In the case of the hopping times, there is a shift in the overall magnitude but in both cases the characteristic hopping times are slow and the overall temperature dependence is similar in the two geometries. The corresponding results for the carrier mobility and carrier concentrations are also very similar. We believe that the difference in is related to impurities, as polaronic carries are strongly pinned by disorder creating variations in hopping times. We have included these new results in the supplementary materials and inserted additional discussion into the main text about the overall reproducibility of the results and the sample dependence. These results speak to the reproducibility of our results and the robustness of our conclusions against sample variations and the size of the air gap.

The reviewer wrote: *4. Do the authors also observe hysteresis in these measurements?*

Our response: We do observe hysteresis in the BDS data for NdNiO₃; however, we decided not to discuss it extensively in the text for brevity, as it has been observed before.

The reviewer wrote: *Other comments: The author should cite previous papers on short range monoclinic distortion in metallic phase of nickelates*

Our response: We thank the reviewer for noting these additional works and we have now cited them in the text.

REVIEWERS' COMMENTS:

Reviewer #1 (Remarks to the Author):

First, I'd like to thank the authors for reading carefully all three reviewers' comments and questions and responding in a detailed manner. They have answered all of my comments and questions to my satisfaction, and quite enjoyed reading their explanations as well. I also appreciate that they did not cut corners on getting new experimental evidence, and even prepared new samples to make more measurements.

The total scattering part is quite important in this paper and demonstrates how a technique like this can reveal new phenomena in an important category of materials, specifically quantum materials such as the nickelates, that would have otherwise been difficult to observe. For this reason, among the presentation of the work itself, I recommend publication in Nature Comm.

Reviewer #2 (Remarks to the Author):

In the response and revised manuscript, the authors have well clarified all the questions from me and other reviewers. I consider that the paper becomes much convincing for readers. So now I can recommend the publication of this paper in Nature Communications in the present form.

One minor comment:

Page 8, line 16: "the motion carriers" will be "the motion of carriers".

Reviewer #3 (Remarks to the Author):

I have reviewed updated version of manuscript and the response letter. I appreciate the authors' efforts to revise the manuscript following the comments of the reviewers. Most of my questions have been addressed adequately. I am recommending publication of this revised manuscript in Nature Communications.

Response to Reviewer 1

The reviewer wrote: *First, I'd like to thank the authors for reading carefully all three reviewers' comments and questions and responding in a detailed manner. They have answered all of my comments and questions to my satisfaction, and quite enjoyed reading their explanations as well. I also appreciate that they did not cut corners on getting new experimental evidence, and even prepared new samples to make more measurements.*

The total scattering part is quite important in this paper and demonstrates how a technique like this can reveal new phenomena in an important category of materials, specifically quantum materials such as the nickelates, that would have otherwise been difficult to observe. For this reason, among the presentation of the work itself, I recommend publication in Nature Comm.

Our response: We again thank the referee for their time in reviewing our work and for their positive recommendation. We also appreciate their comments about our additional efforts in responding to the previous round of reports.

Response to Reviewer 2

The reviewer wrote: *In the response and revised manuscript, the authors have well clarified all the questions from me and other reviewers. I consider that the paper becomes much convincing for readers. So now I can recommend the publication of this paper in Nature Communications in the present form.*

One minor comment: Page 8, line 16: "the motion carriers" will be "the motion of carriers".

Our response: We once again thank the referee for their time in reviewing our work and for their positive recommendation. We also thank them for pointing out this typo, which we have corrected in the resubmitted version.

Response to Reviewer 3

The reviewer wrote: *I have reviewed updated version of manuscript and the response letter. I appreciate the authors' efforts to revise the manuscript following the comments of the reviewers. Most of my questions have been addressed adequately. I am recommending publication of this revised manuscript in Nature Communications.*

Our response: We again thank the referee for their time in reviewing our work and for their positive recommendation.